# Multi-Scale Modelling of Plastic Deformation, Damage and Relaxation in Epoxy Resins

**DOI:** 10.3390/polym14163240

**Published:** 2022-08-09

**Authors:** Julian Konrad, Sebastian Pfaller, Dirk Zahn

**Affiliations:** 1Lehrstuhl für Theoretische Chemie/Computer Chemie Centrum, Friedrich-Alexander-Universität Erlangen-Nürnberg, Nägelsbachstr. 25, 91052 Erlangen, Germany; 2Lehrstuhl für Technische Mechanik, Friedrich-Alexander-Universität Erlangen-Nürnberg, Egerlandstr. 5, 91058 Erlangen, Germany

**Keywords:** epoxy resins, molecular dynamics, constitutive modelling

## Abstract

Epoxy resin plasticity and damage was studied from molecular dynamic simulations and interpreted by the help of constitutive modelling. For the latter, we suggested a physically motivated approach that aims at interpolating two well-defined limiting cases; namely, pulling at the vanishing strain rate and very rapid deformation; here, taken as 50% of the speed of sound of the material. In turn, to consider 0.1–10-m/s-scale deformation rates, we employed a simple relaxation model featuring exponential stress decay with a relaxation time of 1.5 ns. As benchmarks, deformation and strain reversal runs were performed by molecular dynamic simulations using two different strain rates. Our analyses show the importance of molecular rearrangements within the epoxy network loops for rationalizing the strain-rate dependence of plasticity and residual stress upon strain reversal. To this end, our constitutive model reasonably reproduced experimental data of elastic and visco-elastic epoxy deformation, along with the maximum stress experienced before fracturing. Moreover, we show the importance of introducing damage elements for mimicking the mechanical behavior of epoxy resins.

## 1. Introduction

Epoxy resins and their composites with fibers and particles have become indispensable construction materials for a continuously growing range of industrial applications. These range from using reinforced polymers in lightweight, mechanically resilient devices to use as versatile repair materials [1,2,3]. Due to the enormous engineering efforts invested within many decades, state-of-the-art epoxies currently represent practically ideal chemical bonding and unreacted moieties, with thermosetting polymers amounting to less than 1%. Despite this near 100% degree of crosslinking, epoxy resins feature amorphous network structures with only limited ordering at the 1-nm-length scale [4,5,6,7]. This implies an interplay of local chemistry and more global material properties, the understanding of which is an ongoing challenge to both experiments and theories. 

While we are still in the beginning stage of this endeavor, molecular simulation approaches recently showed significant advances in elaborating realistic models of crosslinked polymers, such as epoxy resins. Pioneering studies dating back to more than ten years ago reasonably reproduced elastic moduli and bulk density, albeit reaching hardly more than 70% of crosslinking in the underlying network models [5,8,9]. On this basis, the fundamental aspects of epoxy characteristics beyond the elastic regime, such as glass transition points and yielding, could be assessed at least from a qualitative point of view [10]. More recently, however, improved crosslinking algorithms paved the way to models with up to 99%-reacted epoxy resins [4,11,12]. On this basis, we can reproduce key experimental data, such as the degree of curing, heat of polymerization reactions, elastic moduli, yield stress, and ultimate stress before fracturing [13]. 

At length scales larger than ~5 nm, epoxy resins display structural homogeneity and thus isotropic elasticity behavior. In turn, anisotropic behavior is observed upon deformation beyond the yielding point. Visco-elastic modes and plastic deformation behavior account for manifold processes in the polymer. These may span from twisting and sliding of individual loops within intact networks of covalently linked epoxy moieties to bond-reorganization and finally the dissociation of covalent bonds [13]. Thus, insights at the 0.1 to 1 nm scale are crucial for an in-depth understanding of plasticity, damage, and fracturing.

Based on reactive atomic interaction models, we recently enabled molecular dynamic simulations to study thousands of molecule-sized simulation cells of bisphenol-F-diglycidyl-ether, BFDGE, epoxy monomers and 4,6-diethyl-2-methylbenzene-1,3-diamine, DETDA, and linkers [4,13]. Although the molecular simulation models were limited to the study of incompletely reacted networks created from brute-force approaches, the key motivation for our simulation system was in its closeness to the experimental formulation of the resin [4]. The simulation methods and models were rigorously benchmarked to ensure realistic setups that reproduced the experimental heat of curing reactions, structural data (density, 98% crosslinking), elastic properties, yielding, and fracture behavior upon tensile deformation [4,13]. We thus propose that our BFDE-DETDA system is a particularly robust starting point for investigating complex setups of mechanical testing.

From the perspective of epoxy resin application as a building part, re-occurring loading and unloading is of immense importance. In this regard, a plethora of approaches has been published. Recent studies have focused on the constitutive modelling of thermosetting polymers regarding temperature-dependent and visco-elastic effects [14,15], as well as the multiscale treatment of elastoplasticity [16] of neat polymers and the constitutive descriptions of their composites [17]. Furthermore, sophisticated constitutive models are available for the curing process, e.g., [18]. However, many models at hand suffer from the fact that they require a large number of material parameters that have to be calibrated against either experimental results or numerical findings obtained from sources such as molecular dynamic simulations. This usually yields an ill-posed problem with a very limited physical meaning of the resulting parameter set. To overcome this, in this paper, we propose a strategy to derive a constitutive model of plasticity and damage using a minimal set of physically motivated elements, whereas both parameterization and interpretation stem from explicit molecular dynamic simulations. In what follows, we outline our approach by means of strain reversal during cyclic deformation.

## 2. Models and Methods

Constitutive models of material deformation take use of intuitive mechanical elements to describe the loading and relaxation behavior. For this, simplified basic elements, such as Hookean springs, St. Venant, and Newtonian elements have been combined into a network of serial/parallel arrangements [19,20]. The outline of such mechanical models may follow two different strategies; namely, (i) aiming at the prediction of stress–strain diagrams as functions of strain rates by means of mathematical optimization, and (ii) the rationalization of the underlying mechanical processes by well-defined physical concepts. These two aspects do not necessarily comply with each other. For (i), researchers commonly use a large number of mechanical elements to achieve mathematical descriptions with high accuracy. This gives rise to a large set of adjustable variables typically generated from directly fitting the constitutive models to stress–strain data obtained from cyclic deformation runs. In turn, for (ii), to interpretate the constitutive models, it is desirable to focus on a minimalistic setup of mechanical elements, such that each element reflects a particular mechanistic aspect of the system.

In this study, we aimed at the formulation of an irreducible set of mechanical elements to describe the cyclic testing of epoxy resins. For this, we revisited our earlier work on EPON-DEDTA resin deformation and fracture [13] and extended the underlying simulation protocols to strain reversal. In favor of creating a physics-driven mechanical model, we suggested strain-rate-dependent deformation as an interpolation scenario between two limiting cases: the stress–strain diagram at a vanishing strain rate and very fast deformation (e.g., half of the speed of sound in the material). In turn, we reserved the molecular dynamic simulation data of the strain reversal runs for benchmarking the constitutive model.

The standards for the mechanical testing were adopted from our earlier study described in detail in [13]. In brief, the molecular dynamic simulations were based on a time step of 1 fs and used a thermostat to maintain 300 K. Before each molecular dynamics (MD) time step, the molecular models were scaled along the pulling direction according to the corresponding strain rates. Volume relaxation was allowed from a two-dimensional barostat algorithm that imposed 1 atm normal to the direction of the strain application.

The two stress–strain profiles used as inputs in our constitutive modelling are shown in Figure 1a. The underlying molecular dynamic model of the 3D periodic epoxy bulk, as reported in [13], featured an initially cubic cell with *l*_0_ = 8.4-nm-long dimensions. Elongation *s* of this model thus referred to a technical strain of *ε* = *s*/*l*_0_. For the limiting case of a vanishing strain rate $\sigma \left(s;\;\dot{s}\to 0\right)$, we took use of the pull and hold runs described in detail in our earlier study [13]. Therein, a series of snapshots from the tensile testing was subjected to 10-ns-scale relaxation runs at constant elongation. This led to time-dependent profiles of stress relaxation $\sigma \left(t\;;\;s=const.\right)$ that were fitted as an exponential decay. Consequently, extrapolation to t→∞ in $\sigma \left(t\;;\;s=const.\right)$ was used as the quasi-static estimate of $\sigma \left(s;\;\dot{s}\to 0\right)$. Moreover, the underlying relaxation time of τ = 1.5 ns from the exponential fits indicated the role of local molecular rearrangements for epoxy deformation at a non-zero strain rate [13]. 

To account for very rapid deformation, we defined the upper boundary of deformation speed as half of the speed of sound in the material. Using the same molecular dynamic model as reported in [13], we found the stress profile $\sigma \left(s;\;\dot{s}=\frac{1}{2}v_{sound}\right)$ to essentially follow an elastic-type behavior until there was an abrupt fracture at an ultimate stress of *σ_max_* = 185 MPa (Figure 1a). We took this as the upper limit of stress at which epoxy fracturing occurred as a concerted cleavage of bonds. At a vanishing strain rate, we found gradual cavitation and crack propagation and an ultimate stress of 84 MPa for $\dot{s}=0$, which was in excellent agreement to the experiments of Littell et al. [21].

Our constitutive model, as illustrated in Figure 1b, consisted of an elastic module (left) represented by a Hooke model and an elasto-visco-plastic module (right), described by the Perzyna model. This, in turn, comprised another serial arrangement of the Hooke model and visco-plastic Bingham model. This construction stemmed from a two-step approach. The left-hand side of the diagram refers to the limiting case of a vanishing strain rate. Therein, the elastic-to-plastic deformation regime up to *s_crit_* was described by a parallel set of Hookean and St. Venant elements, the constants of which were directly taken from the $\sigma \left(s;\;\dot{s}\to 0\right)$ profile in Figure 1a. The Hookean element *K_el_* = 31 MPa/nm reflected the elastic properties, whereas the stress reduction during plastic deformation was represented by the spring element Δ*K_pl_* = 25 MPa/nm. The St. Venant delimiter for the onset of plastic deformation modes was derived as *σ_pl_* = +31 MPa (Figure 1a). To account for the entire deformation and fracture profile, we furthermore introduced a damage element. For this, the linear decline of stress upon elongation beyond *s_crit_* was directly adopted from the $\sigma \left(s;\;\dot{s}\to 0\right)$ profile of [13].

By introducing a damping element parallel to the branch mimicking the plastic aspect of the $\sigma \left(s;\;\dot{s}\to 0\right)$ profile, we accounted for deformations at a non-zero rate. Within the constitutive model, pulling at infinite speed referred to the linear regime of the stress profile $\sigma \left(s;\;\dot{s}=\frac{1}{2}v_{sound}\right)$ in Figure 1a, whereas finite deformation rates accounted for the damping element τ (Figure 1b). This damping element described the material’s behavior during pull and hold scenarios, which we already identified as the exponential relaxation of momentary stress towards the stress profile of the quasi-static tensile testing. In other terms, the damping constant 1/τ with τ = 1.5 ns was directly adopted from our relaxation study reported in [13].

To this end, the presented constitutive model featured only one newly fitted parameter; namely, the maximum stress experienced before the *concerted* bond rupture. This limit was only relevant for very rapid pulling, with an $\dot{s}$ much larger than 10 m/s. The deformation rate for assessing the upper stress delimiter of *σ_max_* = 185 MPa was somewhat arbitrarily chosen as $\dot{s}$ = 840 m/s (which was roughly half the speed of sound in the resin). However, in our earlier study, a pulling rate of $\dot{s}$ = 84 m/s led to an ultimate stress of about 180 MPa [22], thus indicating a rather insensitive identification of the *σ_max_* parameter for $\dot{s}=0.05-0.5$ times $v_{sound}$.

The constitutive model was thus implemented as a set of differential equations that were numerically integrated. For this, robust numerical accuracy was found for a time step of Δ*t* = 0.001∙$8.4\;\mathrm{nm}/\dot{s}$. For each time increment, *s* was propagated according to: (1)$$s\to s+\dot{s}\cdot ∆t-\Delta s_{damage}\;\;\mathrm{with}\;\;\Delta s_{damage}=max\left\{0,\frac{s-s_{crit}}{s_{sep}-s_{crit}}\right\}$$
where the damage model only affected elongations beyond *s_crit_* = 10 nm and was inactive during the cyclic deformation runs up to 8.4 nm, which we discuss later. Next, we considered (i) the change in momentary stress from elongation/strain reversal by Δ*s* and (ii) the stress relaxation according to the damping term. Thus,
(2)$$\sigma \left(t\right)\to \sigma \left(t+∆t\right)=\sigma \left(t\right)+K_{el}\cdot \dot{s}\cdot ∆t\;-\left[\sigma \left(t\right)-\sigma \left(s\;;\dot{s}\to 0\right)\right]\cdot \frac{∆t}{\tau }$$

Accordingly, we assumed the *immediate* response of the material to increments of the deformation Δs as ideally elastic. In turn, the time-dependence of the *momentary*
*stress σ*(*t*) was given by the first-order kinetics of its decay towards the quasi-static limit. The latter depended on the *overall* deformation *s*. 

To benchmark the constitutive modelling of tensile deformation/strain reversal cycles, we took use of our previously elaborated simulation system featuring a 3D periodic bulk model of epoxy resin [13]. Thus, all molecular interaction potentials, the molecular dynamic protocols, and the simulation cell were fully adopted from [13]. The starting system of 2048 EPON and 1024 DETDA, linked into a network with a 98% curing degree, reflected a cubic cell of 8.4-nm-long dimensions. In line with our earlier study, for the present study on deformation cycles, we applied elongations at a speed of $\dot{s}$ = 0.84 and 8.4 m/s, which referred to technical strain rates of $\dot{\varepsilon }$ = 10^8^ and 10^9^ s^−1^, respectively. 

## 3. Results

Our constitutive model was first applied to tensile testing and epoxy fracture runs using different strain rates in line with previous molecular dynamic runs [13]. The latter referred to deformation at a speed of $\dot{s}$ = 0.84 and 8.4 m/s, respectively. Both molecular dynamic runs thus offered direct benchmarking of the constitutive model, which, in turn, was fitted to the limiting scenarios of $\dot{s}$ = 0 and $\dot{s}\to \infty ,\;\mathrm{respectively}$. In Figure 2, we show the momentary stress as a function of elongation $s=\dot{s}\cdot t$ as predicted from the constitutive model and the corresponding molecular dynamic data. Despite the very limited number of mechanical elements used, our simplistic model at least qualitatively reproduced the stress profiles of tensile testing runs, including plastic deformation and fracturing. From a more rigorous quantitative viewpoint, however, we found the constitutive model to overestimate the role of the deformation speed. This particularly applied to the cavitation and fracture processes, namely, the stress profiles at *s* > 10 nm. On one side, this may have been related to our rather simple damage model, which suggests a linear decline of the elastic constants after pulling beyond *s_crit_*. However, our constitutive model also relied on a rather strong simplification of time-dependent relaxation.

To further elucidate this issue, the constitutive model was applied to a series of tensile testing and strain reversal runs. In parallel setups, deformation *s* up to various elongations *s**_max_* was performed, followed by a full reversal to *s* = 0. For this, *s**_max_* was chosen as 2, 8, 20, and 100% of the size *l*_0_ = 8.4 nm of the pristine epoxy model. This selection of maximal deformation *s**_max_* < *s**_crit_* was chosen in such a way that our epoxy system remained in the plastic deformation regime. In other terms, the damage model used for describing cavitation and fracture did not apply, and the cyclic deformation runs instead put a focus on the visco-elastic relaxation and plastic behavior.

The stress profiles for two sets of cyclic deformation runs are illustrated in Figure 3. To benchmark the constitutive model, molecular dynamic simulation runs were performed using the same selection of *s**_max_* and $\dot{s}$. From this, we found that the overall profile of the momentary stress was reasonably well reproduced, at least from a qualitative viewpoint. For a more quantitative comparison, we assessed the momentary stress as observed upon full reversal to *s* = 0 (hence imposing a full cycle of $s=0\to s_{max}\to 0$). A comparison of the residual stress *σ*(*s* = 0) as observed after a full reversal of the deformation is provided in Table 1. The momentary stress calculated from molecular dynamic runs was subject to fluctuations that gave rise to ±5 MPa error margins. As a local indicator of plastic deformation, we monitored the root-mean-square deviation of the atomic positions ${\vec{r}}_{i}\left(s=0\right)$ of the pristine epoxy model as compared to the reversed system, ${\vec{r}}_{i}\left(s=0\;\to \;s_{max}\to 0\right)$.

While the agreement of the constitutive model and molecular dynamic data appeared promising for the last stage of deformation reversal, we found larger discrepancies for the onset of the reversal runs. This led to a quite drastic mismatch in predicting the ‘relaxed state’; that is, the residual deformation after reversing to zero stress, namely $s=0\to s_{max}\to s_{\sigma =0}$. In Table 2, we show $s_{\sigma =0}$ as predicted from the constitutive model along with the corresponding values observed from the momentary stress profiles of the molecular dynamic runs.

$s_{max}=0.17\;\mathrm{nm}$$s_{max}=0.67\;\mathrm{nm}$ Apart from such benchmarking, the deformation reversal runs based on molecular dynamic simulations also offered mechanistic insights that guided appropriate extensions of the constitutive model. To this end, we studied the shape changes of the molecular simulation cells during the deformation cycle. As a measure to tapering, we monitored the momentary cross-section area *A*(*s*), taken as normal to the elongation *s*. This was contrasted to the cross-section *A*_0_ of the pristine model. In Table 1 and Table 2, the relative change of (*A*(*s*) – *A*_0_)/*A*_0_ is indicated. While our epoxy models experienced tapering during deformation, upon full reversal runs of $s=0\to s_{max}\to 0,$ we found only negligible differences of cell shapes before and after the deformation cycle. This was in line with our earlier study that showed the volume conservation of the molecular simulation system up to the onset of cavitation and fracture to be near *s* = 10 nm when experiencing ultimate stress [13]. Our constitutive model approach using a single mechanical element readily described this single-mode, volume-conserved deformation. We thus propose that our constitutive model could only weakly benefit from extensions to multiple Hookean elements (e.g., for uniaxial and volumetric terms).

A more local picture of plastic deformation may be derived from the root-mean-square-deviation (*rmsd*) of the molecular simulation systems during cyclic deformation. These are indicated for *s* = 0 and *s* = $s_{\sigma =0}$ in Table 1 and Table 2, respectively. Moreover, in Figure 4, we show a series of snapshots from a cyclic deformation run. 

From this, the heterogeneous nature of epoxy resin loops and nodes was clearly demonstrated. Local reorganization gave rise to rather large atomic displacements with *rmsd* values reaching up to 3 nm. In turn, the degree of crosslinking was observed to reduce only by 0.8%; at the end of the deformation cycle, our simulation cell still showed an epoxy network of 97% crosslinking. 

While the net loss of network connectivity represented a plastic deformation mechanism, network re-arrangements by the twisting and sliding of loops can be, at least to some extent, attributed to visco-elastic modes. This interplay of modes was further complicated by the observation of bond re-organization, thus showing the cleavage and rebuilding of new polymer loops. For the latter type of network rearrangement, we assumed a strong dependence of the deformation rates: the slower the deformation, the more time was allowed for such extended relaxation. For example, we observed bond-reorganization only for the slower deformation rate, amounting to 0.2% of all crosslinks, whilst no such process was identified for the faster deformation cycle.

All these types of atomic displacements were thus driven by momentary stress and strain-rate-dependent relaxation into alternative molecular configurations. In Figure 5, this is exemplified from comparing the extent and distribution of *rmsd* values as a function of the elongation speed $\dot{s}$ for snapshots of the molecular dynamic runs at *s_σ_*_=0_.

In more general terms, the overall manifold of possible atomic arrangements was related to a very complex energy landscape of high dimensionality. This featured manifold minima and, in line with the amorphous nature of the epoxy network, we suggest many of these energy minima were equivalent to that of the molecular system before deformation. Consequently, alternative network arrangements may have represented stable configurations that did not experience driving forces (i.e., *σ* = 0) to restore the pristine system. We suggest this as the molecular scale mechanism of plasticity.

To underpin this picture, we performed additional molecular dynamic runs (i) at the constant *s* = 0 after completion of the deformation cycle with *s**_max_* = 8.4 nm. Based on a nanosecond-scale relaxation at 300 K, we found the momentary stress to exponentially decay to a non-zero residual stress of $\sigma \left(t\to \infty \right)$ = −17 and −20 MPa, for snapshots taken from cyclic deformation by $\dot{s}$ = 0.84 and 8.4 m∙s^−1^, respectively. Likewise, (ii) free relaxation at 1 atm and 300 K of the snapshots from cyclic deformation by $\dot{s}$ = 0.84 and 8.4 m∙s^−1^ taken for *s_σ_*_=0_ showed an exponential relaxation of *s*(*t*), in favor of the non-zero residual deformation *s*(*t*→∞) = 4.8 and 5.2 nm, respectively.

These findings contradicted the constitutive model that predicts *s_σ_*_=0_ = 0 for vanishing deformation rates. Consequently, we propose that a physics-driven approach to better account for plasticity could avoid additional damping and Hookean mechanical elements. Instead, we advocate for the development of a model that can describe an offset ∆*s_plastic_* to elongation *s* because of plastic deformation. We attempted to formulate this in analogy to the damage model for ∆*s_damage_* that we suggested in Equation (1); however, we found only unsatisfactory changes in the predicted stress profiles. Therefore, more than the simple linear model that we tested in the present study is required, and we leave this issue for future research. Such developments can improve the description of the strong change of momentary stress during the initial stage of visco-elastic deformation and at the beginning of the reversal runs.

## 4. Discussion and Conclusions

The bridging of length and time scales from the MD simulations (micro to meso) to experiments (macro scale) represented an immense challenge. The presented constitutive models reached the macro scale, but also showed the need of improvements for assessing the time scales inherent to common material testing experiments. 

Our constitutive model was formulated as a minimalistic, irreducible set of well-defined mechanical elements. A comparison with the momentary stress profiles from molecular-dynamic-reference simulation runs revealed some shortcomings in the predictive modelling at its current stage. From a mathematical viewpoint, the use of more mechanical elements has enabled more accurate fitting [23,24,25,26,27,28]. However, the larger the number of parameters, the more challenging the physical interpretation [27]. In the present contribution, we therefore did not further expand the constitutive model, but instead analyzed the physical foundations of the observed discrepancies. 

From discriminating different aspects of the momentary stress profiles, we identified the description of residual deformation after cyclic tensile testing as a fundamental problem to constitutive modelling. The widely used approach of introducing further damping elements may improve the mathematical reproduction for a given range of strain rates, but it cannot fix the misconception identified for the limiting case of the vanishing speed of deformation. 

The presented constitutive model was nevertheless quite helpful for the interpretation of our molecular dynamic runs. While the elastic and visco-elastic aspects of epoxy deformation and relaxation are well described by mechanical elements, the difference of the predicted stress diagrams regarding the molecular dynamics data may be directly attributed to the role of plasticity. In other terms, we suggest such a comparison as a quantification of the contributions from viscous and residual modes, providing at least some guidance to tackle the immense complexity of the extended structural reorganization of the epoxy network observed from molecular dynamic simulations.

While the present study focused on pure epoxy resins, the elaborated modelling and simulation approaches appeared quite transferrable. Provided that reference molecular dynamic runs are available, the described designing of constitutive models can also qualify for polymer composites involving particles and fiber components. This would allow the rationalization of reinforced polymers by comparing the individual mechanical elements. To this end, we suggest the present study as a reference point and envisage composite analyses for future studies.

## Figures and Tables

**Figure 1 polymers-14-03240-f001:**
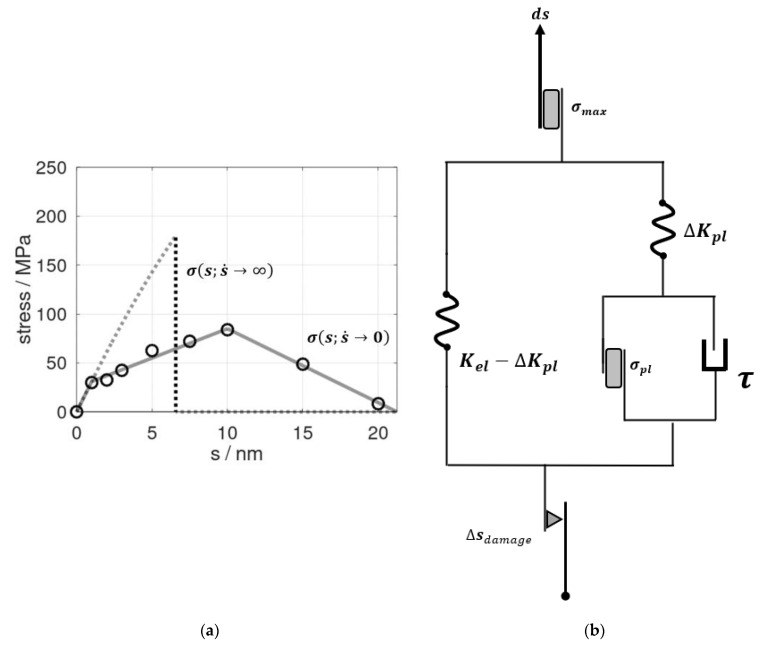
(**a**) Stress profiles as functions of the elongation ***s*** of the epoxy model. The dots refer to data from molecular dynamic simulations combined with extrapolation analyses for t→∞ of stress relaxation at constant *s* as adopted from [13]. On this basis, $\sigma \left(s;\;\dot{s}\to 0\right)$ was mimicked by a tri-linear profile discriminating the elastic/visco-elastic regime (up to *s* = 1 nm), plastic deformation (1 nm < *s* ≤ *s_crit_* = 10 nm), and cavitation/fracture (10 nm < *s* ≤ *s*_sep_ = 21.2 nm), as shown by the solid lines. In turn, the stress induced by rapid pulling (dashed curve) was modelled by an effective elastic profile for which the spring constant reflected Young’s modulus up to *s* = 1 nm. At a maximum stress of $\sigma_{max}\left(s=6.5\;\mathrm{nm}\;;\;\dot{s}=\frac{1}{2}v_{sound}\right)=$ 185 MPa, an abrupt failure was implemented according to molecular dynamic simulations performed at a pulling rate of 840 m∙s^−1^ (~50% of the speed of sound in the bulk resin). (**b**) Concept of an elastoplastic model to mimic the change of stress d*σ* in response to increments of the elongation d*s*. At the vanishing strain rate $\sigma \left(s;\;\dot{s}\to 0\right)$, a parallel setup of Hookean and St. Venant elements applied. In turn, time-dependent effects were described by a damping element τ, which was connected parallel to the branch describing plastic deformation in the $\sigma \left(s;\dot{s}\to 0\right)$ model. For rapid pulling, the stress maximum *σ_max_* was observed near the speed of sound. This was imposed by a St. Venant element (top). Moreover, to describe the decline of the spring constants upon elongation beyond *s* = *s_crit_*, we introduced a damage element Δ*s_damage_* (bottom). This reduced the effective elongation of the (elastoplastic) springs to *s* – Δ*s_damage_* in order to mimic a linear decrease of the stress during cavitation and final rupture at *s* = *s*_sep_.

**Figure 2 polymers-14-03240-f002:**
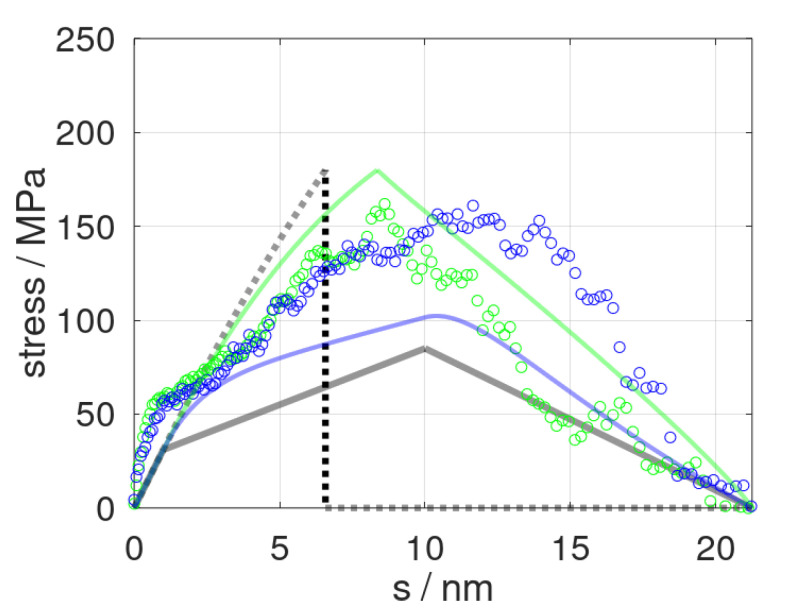
Stress profiles as obtained from molecular dynamic simulations (dots) and predicted stress $\sigma \left(s\right)$ (solid curves) at strain rates of $\dot{s}$ = 0.84 and 8.4 m∙s^−1^ shown in blue and green colors, respectively. In turn, the delimiters for the constitutive model, $\sigma \left(s;\;\dot{s}=0\right)$ and $\sigma \left(s;\;\dot{s}\to \infty \right)$, are indicated by grey lines and dots, respectively.

**Figure 3 polymers-14-03240-f003:**
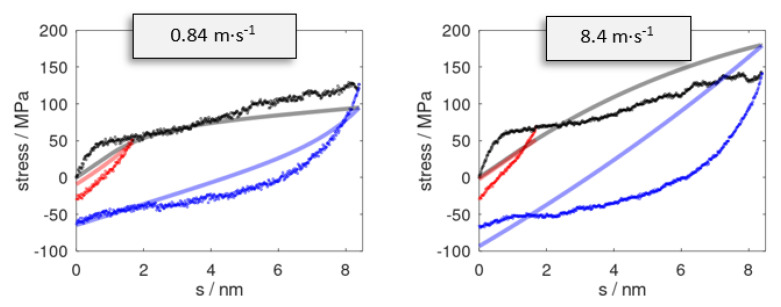
Stress profiles as observed for pulling (black) up to s = 8.4 nm at strain rates of $\dot{s}$ = 0.84 and 8.4 m∙s^−1^. Snapshots at s = 1.7 (red) and 8.4 nm (blue) were subjected to reverse deformation using the analogous rates $\dot{s}\to -\dot{s}$. Data series from the stress prediction model and molecular dynamic simulations are indicated by solid curves and dots, respectively.

**Figure 4 polymers-14-03240-f004:**
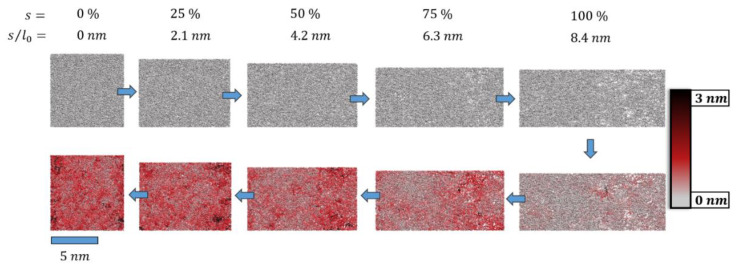
Snapshots of the pristine epoxy model (top left, *l*_0_ = 8.4 nm) before and after deformation by *s* = f∙*l*_0_ (upper panel, $\dot{s}$ = 0.84 m∙s^−1^). After tensile deformation to *s_max_* = 8.4 nm, the system was compressed (lower panel, $\dot{s}$ = −0.84 m∙s^−1^) back to *s* = 0 (lower left). For ideal elastic deformation, the column-wise pairs of snapshots with identical f = 0, 25, 50, 75 and 100% would be identical. In turn, the deviations of atomic positions ${\vec{r}}_{i}\;$ were taken as an indicator of plastic deformation. This is highlighted by a color code to illustrate the root-mean-square-deviation (*rmsd*) of ${\vec{r}}_{i}\left(f,\;\mathrm{pull}\right)-{\vec{r}}_{i}\left(f,\;\mathrm{reversed}\right)$ for each atom *i*. The twisting and sliding of polymer loops upon deformation (far beyond the elastic regime) and strain reversal led to local rearrangements that involved displacements of up to *rmsd* = 3 nm.

**Figure 5 polymers-14-03240-f005:**
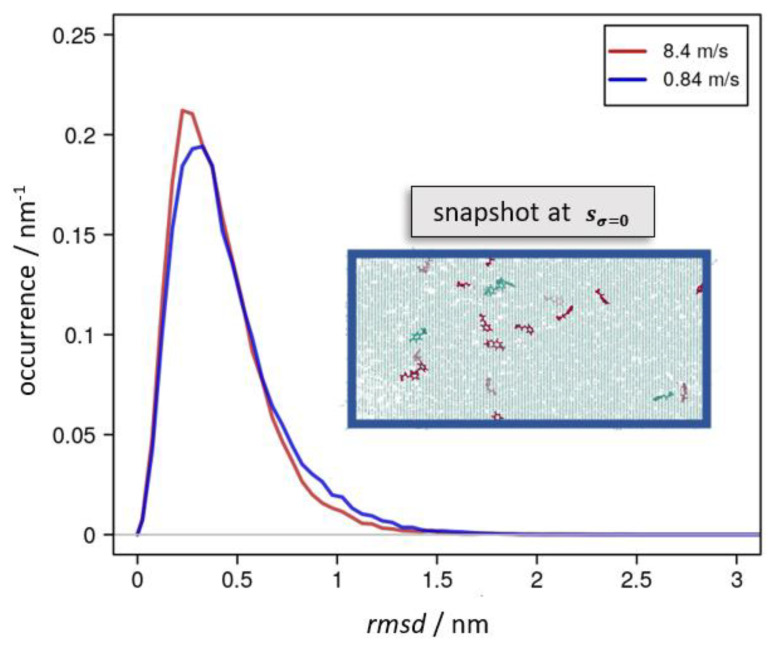
Occurrence profile of the *rmsd* taken for snapshots taking deformation up to 8.4 nm and the corresponding reversal run. For this, configurations of equal elongation *s* were chosen according to *s* = *s_σ_*_=0_ = 6.2 and 6.1 nm, for which the momentary stress reached zero during the reversal runs using 8.4 m∙s^−1^ (red curve) and 0.84 m∙s^−1^ (blue), respectively. For the latter run, the inset shows the epoxy network highlighting EPON moieties that permanently lost one of its links in the red color. Likewise, the green color indicates EPON species that cleaved and rebuilt covalent bonds to the DEDTA linkers.

**Table 1 polymers-14-03240-t001:** Residual stress, change of the simulation cell (cross-section area A normal to the applied elongation *s*), and atomic displacements (*rmsd*) for cyclic deformation runs, as observed from molecular dynamics and constitutive modelling (data denoted in square brackets).

s= $0\to s_{max}\to 0$	($\dot{s}=8.4\;;\;0.84\;m\cdot s^{-1}$) $\sigma_{residual}\;/\mathrm{MPa}$	($\dot{s}=8.4\;;\;0.84\;m\cdot s^{-1}$) $\left(A\left(s=0\right)-A_{0}\right)\;{/A}_{0}$	($\dot{s}=8.4\;;\;0.84\;m\cdot s^{-1}$) rmsd/nm
$s_{max}=0.17\;\mathrm{nm}$	−4; −1 [0 ;0]	0; 0 [n/a]	0.9; 0.9 [ n/a ]
$s_{max}=0.67\;\mathrm{nm}$	−8; −8 [0 ;−8]	0; 0	1.0; 1.1
$s_{max}=1.68\;\mathrm{nm}$	−29; −29 [−2;−29]	0; 0	1.3; 1.6
$s_{max}=8.4\;\mathrm{nm}$	−66; −63 [−93;−63]	0.02; 0.01	2.7; 2.8

**Table 2 polymers-14-03240-t002:** Shape changes of the simulation cell as observed from the analyses of the strain reversal runs starting from a variety of epoxy deformation scenarios *s_max_*. For this, the profiles of the momentary stress were inspected for vanishing residual stress, and the corresponding elongation $s_{\sigma =0}$ was taken as a measure of plasticity. Data are shown for molecular dynamic runs and constitutive modelling (numbers in square brackets).

s= $0\to s_{max}\to s_{\sigma =0}$	($\dot{s}=8.4,\;0.84\;m\cdot s^{-1}$) $s_{\sigma =0}\;/\mathrm{nm}$	($\dot{s}=8.4,\;0.84\;m\cdot s^{-1}$) $\left(A(s_{\sigma =0})\;-\;A_{0}\right){\;/\;A}_{0}$
$s_{max}=0.17\;\mathrm{nm}$	0.03; 0.01 [0 ;0]	0; 0 [n/a]
$s_{max}=0.67\;\mathrm{nm}$	0.1; 0.1 [0 ;0]	0; 0
$s_{max}=1.68\;\mathrm{nm}$	0.6; 0.8 [0 ; 0.2]	0.02; 0.01
$s_{max}=8.4\;\mathrm{nm}$	6.2; 6.1 [3.0 ; 4.1]	0.10; 0.08

## Data Availability

All key data is presented in the manuscript. Molecular Modelling inputs can be made accessible upon request to the authors.

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
