# Peer review of "Multi-Scale Modelling of Plastic Deformation, Damage and Relaxation in Epoxy Resins"

_polymers, 2022, doi:10.3390/polym14163240_

Round 1

Reviewer 1 Report

An interesting study was conducted on the Multi-scale Modelling of plastic Deformation, Damage and Relaxation in Epoxy Resins. However, some basic information is missing or insufficient. It is suggested that the authors consider the following major suggestions to significantly improve the quality of papers to meet the publishing requirement. 

1# In the abstract, please provide some research results of this paper, such as simulation and analysis of plastic deformation damage and relaxation of epoxy resins. In addition, the authors should indicate the accuracy of the molecular dynamics model and its comparison with the experimental results. Some research and application background of epoxy resin should be given at the beginning of the abstract.

2# The introduction writing lacks the necessary information. The specific comments are as follows.

1) The performance, advantages and application fields of epoxy resin as indispensable engineering materials should be analyzed in detail in the first paragraph. Epoxy has the advantages of simple processing and forming, high production efficiency, high mechanical properties and lower viscosity etc., which has been widely used as engineering materials, such as resin matrix in FRP, adhesives, fillers, coatings and so on. Please refer to the following latest research on performance, advantages and application fields of epoxy resin. Durability applications in complex environments: Composite Structures, 2020; 246: 112418. Marine application: Polymers, 2021, 13:154.  Resin matrix in FRP and anchorage fillers: Materials and Structures, 2020, 53: 73.

2) The analysis results on molecular dynamics model from the others’ should be summarized to provide the support for the present investigation. At the same time, some key simulation results and latest research progress of epoxy or FRP should be further obtained to provide a basic understanding for readers.

3) As a simulation research, the accuracy of the model is critical. The accuracy of the model should be further verified by the experimental results. Therefore, with regard to plastic deformation, damage and relaxation of epoxy resins, the authors should summarize the relevant experimental research results to verify the present model. For the plastic deformation, damage and relaxation of epoxy resin and its composites with or without the loading, please refer to the research (Guo R and Xian GJ,   Abdulkhadar UM and ShivakumarGouda PS,   He H, Chen W and Yin ZY)

4) In the last paragraph of the introduction, the authors should further highlight the main contributions and innovations of this paper by summarizing the research work of others.

3# Please improve the picture quality and standardability of the current paper. In addition, please remove the redundant scales on the top and right (such as Figure 1a). In addition, Figure 1a should be named as Figure 1 and Figure 1b should be named as Figure 2.

4# In part 2 (Models and Methods), it seems that the logic of writing is not clear, which is easy to make the readers and reviewers confused. A suggestion is that the authors can provide a flow chart of modeling and briefly describe the several steps of model establishment.

5# In the part of results, how to verify the accuracy of the model? Can the mechanical property parameters output from the model be related to the experimental results? This is a very critical problem. Even if the quantitative relationship between simulation and experimental results cannot be established directly, qualitative analysis and description are necessary to analyze the rationality and effectiveness of the current model.

6# All table names are too long. It is recommended to make the further refinement. You can add some notes under the table for necessary explanations.

7# Authors adopted the molecular dynamics simulation to establish micro-scale models, and how to further establish the relationship between micro-scale models and meso-scale and macro-scale performances, which will become the groundbreaking research.

8# The authors should rewrite the current conclusion. Firstly, the current conclusion should be refined, including the critical information. The second point is to summarize the research results of this paper. Therefore, the citations of relevant references should be removed.

9# Finally, there are too few references in the list, and the research status and background are not clearly given. Therefore, it is suggested that the authors provide more research work in recent 2-3 years in terms of the current research. 

Author Response

Ref1

An interesting study was conducted on the Multi-scale Modelling of plastic Deformation, Damage and Relaxation in Epoxy Resins. However, some basic information is missing or insufficient. It is suggested that the authors consider the following major suggestions to significantly improve the quality of papers to meet the publishing requirement. 

1# In the abstract, please provide some research results of this paper, such as simulation and analysis of plastic deformation damage and relaxation of epoxy resins. In addition, the authors should indicate the accuracy of the molecular dynamics model and its comparison with the experimental results. Some research and application background of epoxy resin should be given at the beginning of the abstract.

Good hint, we extended the abstract to discuss the agreement with the experiments. Details on epoxy application are discussed in the introduction.

2# The introduction writing lacks the necessary information. The specific comments are as follows.

1) The performance, advantages and application fields of epoxy resin as indispensable engineering materials should be analyzed in detail in the first paragraph. Epoxy has the advantages of simple processing and forming, high production efficiency, high mechanical properties and lower viscosity etc., which has been widely used as engineering materials, such as resin matrix in FRP, adhesives, fillers, coatings and so on. Please refer to the following latest research on performance, advantages and application fields of epoxy resin. Durability applications in complex environments: Composite Structures, 2020; 246: 112418. Marine application: Polymers, 2021, 13:154.  Resin matrix in FRP and anchorage fillers: Materials and Structures, 2020, 53: 73.

We are well aware of the immense manifold of polymer composite applications. However, current MD and constitutive modelling has to approach this field step-by-step. At the time being, we managed to properly model the pure bulk of the discussed epoxy resin. The introduction of fibers and particles, as well as domain boundaries exceeds the scope of the present paper.

2) The analysis results on molecular dynamics model from the others’ should be summarized to provide the support for the present investigation. At the same time, some key simulation results and latest research progress of epoxy or FRP should be further obtained to provide a basic understanding for readers.

While referring to references 1 and 5 for much more detail, we now describe our motivation for the molecular mechanics models more clearly. The corresponding paragraph in the introduction now reads:

Based on reactive atomic interaction models, we recently enabled molecular dynamics simulations to study thousands of molecules sized simulation cells of bisphenol-F-diglycidyl-ether, BFDGE, epoxy monomers and 4,6-diethyl-2-methylbenzene-1,3-diamine, DETDA, linkers [1,5]. While molecular simulation models for far were limited to the study of incompletely reacted networks created from brute-force approaches, the key motivation for our simulation system is in its closeness to the experimental formulation of the resin [1]. Indeed, the simulation methods and models were rigorously benchmarked to ensure realistic setups that reproduce the experimental heat of curing reactions, structural data (density, 98% crosslinking), elastic properties, yielding and fracture behavior upon tensile deformation [1,5]. We thus argue that our BFDE-DETDA system should be a particularly robust starting point for investigating also complex setups of mechanical testing.

3) As a simulation research, the accuracy of the model is critical. The accuracy of the model should be further verified by the experimental results. Therefore, with regard to plastic deformation, damage and relaxation of epoxy resins, the authors should summarize the relevant experimental research results to verify the present model. For the plastic deformation, damage and relaxation of epoxy resin and its composites with or without the loading, please refer to the research (Guo R and Xian GJ,   Abdulkhadar UM and ShivakumarGouda PS,   He H, Chen W and Yin ZY)

We fully agree on the need for good accuracy. This is indeed the reason why we chose for the specific models described in the present work. Regarding polymer density, elastic deformation, degree of curing and heat of formation, we refer to reference [1], see also response to the previous point.

Regarding the maximum stress, we discussed the experiments of experiments of Littell et al, reference 8. In the modelling and methods section, the corresponding paragraph reads:

To account for (very) rapid deformation, we define an upper boundary of deformation speed as half of the speed of sound in the material. Using the same molecular dynamics model as reported in ref. [5], we found the stress profile  to essentially follow an elastic-type behavior until abrupt fracture at an ultimate stress of σmax=185 MPa (fig. 1a). We take this as the upper limit of stress at which epoxy fracture occurs as a concerted cleavage of bonds. Indeed, at vanishing strain rate we found gradual cavitation and crack propagation and an ultimate stress of 84 MPa for , which is in excellent agreement to the experiments of Littell et al [8].

4) In the last paragraph of the introduction, the authors should further highlight the main contributions and innovations of this paper by summarizing the research work of others.

See response to 2)

3# Please improve the picture quality and standardability of the current paper. In addition, please remove the redundant scales on the top and right (such as Figure 1a). In addition, Figure 1a should be named as Figure 1 and Figure 1b should be named as Figure 2.

Thanks for the hint, the figures experienced some formatting problems which were fixed.

4# In part 2 (Models and Methods), it seems that the logic of writing is not clear, which is easy to make the readers and reviewers confused. A suggestion is that the authors can provide a flow chart of modeling and briefly describe the several steps of model establishment.

We understand this issue, and considered the suggestion carefully. In principle, starting the methods section with the scheme in figure 1b may appear more assessable. However, we wish to point out that the constitutive model was derived from physical considerations. This is why we need to discuss some of the MD results first (fig.1a).

5# In the part of results, how to verify the accuracy of the model? Can the mechanical property parameters output from the model be related to the experimental results? This is a very critical problem. Even if the quantitative relationship between simulation and experimental results cannot be established directly, qualitative analysis and description are necessary to analyze the rationality and effectiveness of the current model.

The accuracy of our constitutive models is best assessed from comparison to the MD simulations as shown in figure 2 and 3 and related discussion. For this, a direct comparison is done. To allow comparison to the experiments of Littell et al. (ref. 8), we did an extrapolation to vanishing strain rate, see also response to #2, 3)     

6# All table names are too long. It is recommended to make the further refinement. You can add some notes under the table for necessary explanations.

We prefer the captions as comprehensive as is.

7# Authors adopted the molecular dynamics simulation to establish micro-scale models, and how to further establish the relationship between micro-scale models and meso-scale and macro-scale performances, which will become the groundbreaking research.

We assume that the reviewer refers to the bridging of scales from the MD simulations (micro to meso scale) to the constitutive models that reach out to the macro scale. We now address this point in the conclusion section. The first paragraph of the conclusion now reads:

The bridging of length and time scales from the MD simulations (micro to meso) to experiments (macro scale) represents an immense challenge. The presented constitutive models reach out to the macro scale, but also show the need of improvements for assessing the time scales inherent to common material testing experiments.

8# The authors should rewrite the current conclusion. Firstly, the current conclusion should be refined, including the critical information. The second point is to summarize the research results of this paper. Therefore, the citations of relevant references should be removed.

The conclusion was extended as described in the response to the previous point.

9# Finally, there are too few references in the list, and the research status and background are not clearly given. Therefore, it is suggested that the authors provide more research work in recent 2-3 years in terms of the current research. 

We did a concise account of current research in the particular fields we discussed in the present study. A broad list of polymer applications or the vast number of different epoxy modifications would exceed the scope of the present work.

Reviewer 2 Report

The paper seeks to introduce an approach ‘’ Multi-scale Modelling of plastic Deformation, Damage and Relaxation in Epoxy Resins’’ However, the authors should consider to improve upon the quality to further highlight and emphasis.

1.    Based on the understanding of what should constitutes an abstract, consider adding one or two lines highlighting on the significance of the study at the end of the abstract.

2.    The introduction needs to be improved by relating to the mechanics of the studied materials and their mechanical characteristics. The references to be included are: 10.1007/s10853-022-06994-3, 10.1177/0021998318790093, 10.1016/j.compstruct.2021.114698, 10.1016/j.jiec.2022.06.023 and 10.3390/polym14132662.

3.    Put a space between each variable and its corresponding unit.

4.    In line 64, line 71, line 72 and line 77, kindly refrain from using pronouns as we which appeared here: “we commonly use a large..”. There are numerous and countless “We/Our” in this manuscript that need to be fixed. You need to make the needful adjustments through the whole manuscript.

5.    One standard of writing style should be adopted, uniformity is vital in every scientific writing. If you consider writing fig., it should run through and vice versa but not alternating between fig. and figure.

6.    The results section [line 172 till line 194] defines the testing methodology and nothing yet comes about results. Therefore, these lines have to be moved up to the models section, without causing repetitions in the content presentation.

7.    Figure 2, Figure 3 and Figure 5, all of them were designed using different font sizes & types for the axes titles and numbers. Make sure that one uniform style will be applied for all figures.

8.    Figure 4 wasn’t referred to in the text at all. Also, these grey structures of epoxy don’t show any clear differences between the different types in S %. These figures can be replaced by higher resolution figures in optimized clear sizes, and a full paragraph has to be written referring to that figure and explaining each structure presented.

Author Response

Ref2

The paper seeks to introduce an approach ‘’ Multi-scale Modelling of plastic Deformation, Damage and Relaxation in Epoxy Resins’’ However, the authors should consider to improve upon the quality to further highlight and emphasis.

  1. Based on the understanding of what should constitutes an abstract, consider adding one or two lines highlighting on the significance of the study at the end of the abstract.

Agreed. The abstract now reads:

Epoxy resin plasticity and damage is studied from molecular dynamics simulations and interpreted by the help of constitutive modelling. For the latter; we suggest a physically motivated approach that aims at interpolating two well-defined limiting cases; namely pulling at vanishing strain rate and very rapid deformation; here taken as 50% of the speed of sound of the material. In turn; to consider 0.1-10 m/s scale deformation rates; we employ a simple relaxation model featuring exponential stress decay as a function of time. As benchmarks; deformation and strain reversal runs are performed by molecular dynamics simulations using two different strain rates. Our analyses show the importance of molecular rearrangements within the epoxy network loops for rationalizing the strain-rate dependence of plasticity and residual stress upon strain reversal. To this end, our models reasonably reproduce experimental data of elastic and visco-elastic epoxy deformation, along with the maximum stress experienced before fracture.

  1. The introduction needs to be improved by relating to the mechanics of the studied materials and their mechanical characteristics. The references to be included are: 10.1007/s10853-022-06994-3, Evaluation of synthesized polyaniline nanofibres as corrosion protection film coating on copper substrate by electrophoretic deposition

 10.1177/0021998318790093, Fatigue and tensile behaviors of fiber-reinforced thermosetting composites embedded with nanoparticles

 10.1016/j.compstruct.2021.114698, Developments in polyester composite materials – An in-depth review on natural fibres and nano fillers

 10.1016/j.jiec.2022.06.023 Statistical and qualitative analyses of the kinetic models using electrophoretic deposition of polyaniline

and 10.3390/polym14132662. Influence of Stress Level and Fibre Volume Fraction on Fatigue Performance of Glass Fibre-Reinforced Polyester Composites

We checked the named references, which are all from Zaghloul and coworkers.

We do not see sufficient relevance of these papers to the present work and refrain from artificially increasing citations.

  1. Put a space between each variable and its corresponding unit.

The editorial team already amended this by formatting the numbers and units.

  1. In line 64, line 71, line 72 and line 77, kindly refrain from using pronouns as we which appeared here: “we commonly use a large..”. There are numerous and countless “We/Our” in this manuscript that need to be fixed. You need to make the needful adjustments through the whole manuscript.

This appears to be a matter of personal taste. We feel that a well-assessable flow of reading is ensured.

  1. One standard of writing style should be adopted, uniformity is vital in every scientific writing. If you consider writing fig., it should run through and vice versa but not alternating between fig. and figure.

Journal standards were imposed.

  1. The results section [line 172 till line 194] defines the testing methodology and nothing yet comes about results. Therefore, these lines have to be moved up to the models section, without causing repetitions in the content presentation.

The testing from MD runs was done as part of the present study. We thus prefer to describe the findings from MD within the results section.

  1. Figure 2, Figure 3 and Figure 5, all of them were designed using different font sizes & types for the axes titles and numbers. Make sure that one uniform style will be applied for all figures.

Fixed.

  1. Figure 4 wasn’t referred to in the text at all. Also, these grey structures of epoxy don’t show any clear differences between the different types in S %. These figures can be replaced by higher resolution figures in optimized clear sizes, and a full paragraph has to be written referring to that figure and explaining each structure presented.

At the end of page 8, there was a wrong referencing to figure 3 – which actually meant 4. The discussion of the heterogeneous deformation modes extends an entire page, please refer to pages 8-9.

Reviewer 3 Report

Epoxy resins plasticity and damage are studied using molecular dynamics simulations. All experimental studies carried out in the article were carried out using modern methods for analyzing rheology, kinetics and physical and mechanical characteristics, using methods of statistical processing of experimental data and using complementary methods for studying the properties of the resulting material. But there are a few questions and recommendations:

1. It is necessary to indicate the standards according to which the tests were carried out. This will justify the reproducibility of the obtained results.

2. One of the main problems faced in the development multiscale models of various molecular nanosystems is associated with the need to interface different spatial and temporal scales (from subnanometer level to micro- and macrolevel, from femtoseconds to nano- and microseconds). Pairing levels means sharing critical information between simulation levels to parameterize system models. How is this problem solved in this case? Perhaps it is worth bringing a scheme or algorithm?

3. How can be used the obtained results in practice? In my opinion, the proposed approaches will allow optimizing the parameters of the outer polymer layers near the filler particles, as was shown earlier in the articles of Professor Stukhlyak, using a multi-scale approach to assess the mechanical properties of epoxy composites: https://link.springer.com/article/10.1134/S1029959915010075

4. How does the resulting model account for temperature effects? Is it possible to state that the construction of the initial states of epoxy resins can be reduced to a universal multiscale algorithm?

5. I recommend removed references on literature from the conclusions. Traditionally, a comparative analysis of the results is carried out before conclusions. In the conclusions, the authors report on the NEW results of their scientific and engineering value for world science.

Author Response

Ref3

Epoxy resins plasticity and damage are studied using molecular dynamics simulations. All experimental studies carried out in the article were carried out using modern methods for analyzing rheology, kinetics and physical and mechanical characteristics, using methods of statistical processing of experimental data and using complementary methods for studying the properties of the resulting material. But there are a few questions and recommendations:

  1. It is necessary to indicate the standards according to which the tests were carried out. This will justify the reproducibility of the obtained results.

Thanks for the hint. We added the requested details at the end of the methods section:

The standards for the mechanical testing were adopted from our earlier study described in detail in reference [5]. In brief, the molecular dynamics simulations as based on a time step of 1 fs and use a thermostat to maintain 300 K. Before each MD time step, the molecular models are scaled along the pulling direction according to the corresponding strain rates. Volume relaxation is allowed from a two-dimensional barostat algorithm that imposes 1 atm normal to the direction of strain application.

  1. One of the main problems faced in the development multiscale models of various molecular nanosystems is associated with the need to interface different spatial and temporal scales (from subnanometer level to micro- and macrolevel, from femtoseconds to nano- and microseconds). Pairing levels means sharing critical information between simulation levels to parameterize system models. How is this problem solved in this case? Perhaps it is worth bringing a scheme or algorithm?

There is no simple answer to the question. We however now discuss this point in the conclusion section in more detail. The first part of the conclusion now reads:

The bridging of length and time scales from the MD simulations (micro to meso) to experiments (macro scale) represents an immense challenge. The presented constitutive models reach out to the macro scale, but also show the need of improvements for assessing the time scales inherent to common material testing experiments.   

  1. How can be used the obtained results in practice? In my opinion, the proposed approaches will allow optimizing the parameters of the outer polymer layers near the filler particles, as was shown earlier in the articles of Professor Stukhlyak, using a multi-scale approach to assess the mechanical properties of epoxy composites: https://link.springer.com/article/10.1134/S1029959915010075

This is sure an interesting issue. However, the current models are not yet ready to describe composites at the required level of accuracy. This is why we first elaborated on the pure epoxy system, thus approaching fibers and particles inclusion in future steps.

  1. How does the resulting model account for temperature effects? Is it possible to state that the construction of the initial states of epoxy resins can be reduced to a universal multiscale algorithm?

The MD simulations were done at 300 Kelvin (as now pointed out in the methods section) and the constitutive model thus refers to this temperature. The overall modelling procedure could of course be carried out at different temperatures, provided that the MD runs are re-iterated at the desired temperature.   

  1. I recommend removed references on literature from the conclusions. Traditionally, a comparative analysis of the results is carried out before conclusions. In the conclusions, the authors report on the NEW results of their scientific and engineering value for world science.

We understand this argument, but still wish to describe the context of our findings within the field of other studies.

Round 2

Reviewer 1 Report

Although the authors provided a revision file, most of the comments were not well replied. Specific comments are as follows. 

1) For the Question 1, the information added by the authors does not cover the key points. Some quantitative simulation results should be provided. Readers can also capture some key information in the abstract for the current work. 

2) For the Question 2, this paper focuses on the multi-scale mechanical modelling of epoxy resin. It is essential to summarize the advantages and engineering applications of epoxy resin, such as resin matrix in FRP, adhesives, fillers, coatings and so on. This can make readers better understand the latest progress and application prospects of epoxy resin. At the same time, this summary also makes the research work of this paper more meaningful. Therefore, authors are suggested to refer to the first review and add necessary summaries. For the Question 9, the research background of this paper should be furthermore enriched, especially the advantages, performance, simulation and application of epoxy resin in recent 2-3 years. 

3) For the Question 2, on the accuracy of the model, the authors claimed that the reference [1] was referred. Therefore, what is the innovation of this paper (This is also the problem below: “In the last paragraph of the introduction, the authors should further highlight the main contributions and innovations of this paper by summarizing the research work of others”)? As known, the establishment of the model is important for this paper. On the other hand, the accuracy of the current model cannot completely be verified only through the reference [1]. 

4) For the Question 3, the reviewer did not find some changes in the revised version, such as redundant scales on the top and right in many figures. In addition, there is a lot of content between figure 1a and figure 1b. 

5) For the Question 4, although the authors provided an explanation, the reviewers were still very confused about the writing of models and methods, and it was difficult to understand the logical thought. 

6) For the Question 6, all table names are too long, and it should be placed at the top of the table. 

7) For the Question 7, On the relationship between micro-scale models and meso-scale and macro-scale performances, the relevant description should be placed in the future outlook, which is not the conclusion of this paper.

8) For the Question 8, the writing of conclusions still needs further improvement for critical information. The conclusion is a summary of the present research work. Why are there references from others?

Author Response

We extended the abstract, introduction and discussion+conclusion part according to the reviewer suggestions. Along this line, the former conclusion part was renamed to "discussion and conclusion".

We also amended formatting errors as far as could be seen from our (most recent) MS office version.

The scales indicated on all figures are meaningful and to the best of our knowledge indispensable. If you disagree, please precisely specify at which place you feel that there are redundancies.

Reviewer 2 Report

The authors didn't work on resolving at least half of the issues.

What do you mean by journal standards were imposed?

Comment no.6 still is not fixed and your justification is wrong.

The references provided are recent and related to damage of polymers. These weren't yet added in your revised version.

Kindly work on each of these comments carefully. 

Regards.

Author Response

We are disappointed that the reviewer continues to try making us cite papers of Zaghloul and coworkers - despite obviously poor relevance to the present work. 

We alert the reviewer to not try taking personal benefit from what should be an independent peer-review.  

10.1177/0021998318790093, Fatigue and tensile behaviors of fiber-reinforced thermosetting composites embedded with nanoparticles

10.1016/j.compstruct.2021.114698, Developments in polyester composite materials – An in-depth review on natural fibres and nano fillers

10.1016/j.jiec.2022.06.023 Statistical and qualitative analyses of the kinetic models using electrophoretic deposition of polyaniline

and 10.3390/polym14132662. Influence of Stress Level and Fibre Volume Fraction on Fatigue Performance of Glass Fibre-Reinforced Polyester Composites

Reviewer 3 Report

The authors took into account some of the recommendations, but there are a few more, the consideration of which, in my opinion, will improve the content of the article:

1. I propose to write the title of the article in capital letters:  MULTI-SCALE MODELLING of PLASTIC DEFORMATION, DAMAGE AND RELAXATION IN EPOXY RESINS

2. I recommend to supplement the Introduction. Now only 5 articles have been analysed in the introduction. [5] is an article by the authors of this paper. Therefore, only 4 articles by other authors were considered. This is not enough to justify the relevance of research. There is no description for the extreme necessity of these studies, their value for the development of theory and especially for practice.

3. It is not clear from the article whether it is possible to predict the fracture energy, adhesive properties of the matrix (epoxy resins), like in figs 1,2 in https://www.mdpi.com/2077-1312/8/7/527, obtained by experiments, from molecular dynamics simulations? I would recommend the authors to show practical value of proposed model.

4. If the authors declare the creation of a new model, then it is necessary to compare it with the known ones and show its advantages (it is more accurate, more physically correct, easier for engineering use, etc.). Perhaps it is worth placing a table of comparative analysis of the positive (and negative) features of the proposed model with the known ones at the end of the article. This would show its place in science and practice.

Author Response

We thank the reviewer for the additional constructive comments. We extended abstract, introduction and the discussion and conclusion section accordingly.

Concerning points 3 and 4, we kindly direct the reviewers attention to figure 2 and the related text. The difference between the MD reference and the constitute model appears too large to reliable predict fracture energy. A discussion of possible model extensions - in the context of other models - is given in the discussion and conclusion part.

Round 3

Reviewer 1 Report

No comments

Author Response

Thanks for your support. Following your suggestions, we extended the introduction and now provide a broader discussion of other modelling and simulation studies.